# Can technology-based gait training result in relevant changes of ambulatory function in people with chronic, neurological diagnoses? A longitudinal, cohort study

Eveline S. Graf [1]*, Dino De Bon[2], Johanna Stahl[3], Jürgen Degenfellner[1], Deborah Knechtle[2], Daniel Zutter[4], Florian Liberatore[3], Markus Wirz[1]

**1** School of Health Sciences, ZHAW Zurich University of Applied Sciences, Winterthur, Switzerland, **2** VAMED Rehazentrum Zürich Seefeld, Zurich, Switzerland, **3** School of Management and Law, ZHAW Zurich University of Applied Sciences, Winterthur, Switzerland, **4** Rehaklinik Zihlschlacht, Zihlschlacht, Switzerland

* eveline.graf@zhaw.ch

## Abstract

### Objectives

To determine whether a six-months technology-based gait training results in relevant changes of ambulatory function in a chronic stage after a neurological event. Further, changes in quality of life will be assessed as well as the willingness to pay for technology-based gait training.

### Design

Single-center, longitudinal cohort study.

### Setting

One outpatient center specialized in neurological rehabilitation.

### Participants

Adults with a chronic neurological diagnosis resulting in residual gait impairments.

### Intervention

Six month of technology-based gait training (with Lokomat, Andago, or C-Mill) with a minimal number of trainings of 10 per month.

### Primary & secondary outcome measures

Walking performance assessed with the 10-meter walk test, 6-minute walk test and functional ambulation category; quality of life assessed with the EQ-5D-3L

**Data availability statement:** Data relevant to the study are available from Zenodo at DOI: 10.5281/zenodo.15268425). All author-generated code has been published on Zenodo at DOI: 10.5281/zenodo.15092410.

**Funding:** The funders consulted during the development of the study design to ensure that it met their funding principles. The final decision on the study design was made independently of the funders by the study team. The funders had no role in data collection and analysis, decision to pub-lish, or prepara-tion of the manuscript.

**Competing interests:** The authors have declared that no competing interests exist.

and WHODAS 2.0; predicted market share and willingness to pay assessed with a choice-based conjoint analysis survey and direct question.

## Results

27 participants completed three months while 20 completed six months of training. Comparing variables at baseline (BL) and after three (M3) and six (M6) months of training, both the walking speed in the 10-meter walk test (BL: 0.46m/s, M3: 0.54 m/s M6: 0.57 m/s) and the distance covered in the 6-minute walk test (BL: 149m, M3: 155m, M6: 159m) showed improvements that were below the clinically relevant change. The quality of life did not indicate a change. The direct willingness to pay was at CHF 80 which resulted in an estimated market share of 55% based on the conjoint analysis.

## Conclusion

The observed, small changes in ambulatory function in patients with chronic, neurological impairments and the willingness to pay indicates the need to provide technology-based gait training in an outpatient setting.

## Introduction

Based on evidence from scientific studies, over the previous two decades a change in the standard practice for neurorehabilitation has occurred. A key to successful rehabilitation is motor learning, which is highly influenced by the intensity of training [1,2]. Intensity is the combination of frequency and duration of training, the number of repetitions of the trained task, together with the required effort. For patients with severe functional impairments, it is difficult to reach the necessary intensity of train-ing without assistive technology. Technology-based locomotor therapy is not a new paradigm, but it enables more repetitions and higher training intensity compared to conventional therapy. The technology, such as a rehabilitation robot, trains the move-ments used in everyday life (e.g., walking), even in cases where a high level of sup-port is required. In addition to supporting patient movements, these devices measure various parameters and provide immediate, quantitative feedback to the user and therapist. These measurements allow the training to be monitored and to objectively quantify its progress. In summary, technology-assisted training enables high repeti-tions, trains functional movements, and provides objective feedback – aspects that are highly relevant for motor learning and successful rehabilitation [1].

Technology-based locomotor training is typically initiated soon after a neurolog-ical event such as a stroke. Early commencement of training has been shown to improve outcomes, particularly in combination with conventional physical therapy [3,4]. The major functional improvements following a neurological event occur in the sub-acute phase where inpatient rehabilitation takes place. It is well documented, that significant improvements are possible for up to a period of six months, albeit at a

decreasing rate [5]. To exploit the potential for further functional improvement beyond the period of inpatient rehabilitation, it is necessary to provide appropriate training stimuli in an outpatient setting. The training loads must be adapted to the recovering capacity [6]. In the initial phase after a neurological event, small stimuli might be appropriate, but increased doses of training are required to trigger the intended improvements in the later phases of rehabilitation. This can be achieved by means of technology-based training. Literature supports the view that locomotor function can improve in response to such training, also in the chronic phases [7,8].

Patients often adopt a sedentary and rather inactive lifestyle after a neurological event [9–11]. Prolonged sitting and a sedentary lifestyle negatively impact on general health. The risk of cardiovascular diseases, such as recurrent stroke, is increased [12,13]. Conversely, it has been shown that an active lifestyle and regular physical activity reduce the risk of a variety of diseases, such as obesity, high blood pressure, diabetes, osteoporosis, and colon and breast cancer. Furthermore, regular activity has been determined to have beneficial effects on psychological well-being and quality of life [14–16]. Unsupported gait causes a high physiological load. Walking with reduced body weight and with exoskeletal support results in a higher physiological load than sitting, and is assumed to be intense enough to result in cardio-vascular training [17]. It is speculated that rehabilitation technology can achieve the recommended health-enhancing physical activity levels that are beneficial to psychological and physiological well-being, which have recently also been stated by the world health organization specifically for people with disabilities [18–20].

In Switzerland, there is a trend towards shortened periods of inpatient rehabilitation. The length of time that patients undergo intensive training is correspondingly reducing, with the average length of stay for inpatient rehabilitation being reduced from an average of 27 days in 2002–24 days in 2022 [21]. Many patients are discharged from inpatient rehabilitation as soon as the supported basic activities of daily living are attained. These shortened programs are probably insufficient for patients to fully exploit their rehabilitation potential. Consequently, it is crucial that outpatient facilities offer access to training programs providing intensive training [22]. Financial models also need to be established to determine the financial contribution of health insurance companies to the cost of such training programs.

In summary, technology-based locomotor training in an outpatient setting for people with severe impairments, through offering the opportunity for continued health-enhancing physical activity, may have the potential to achieve long-term improvement in recovering the functions of everyday life. However, scientific evidence on this subject is lacking. Investigation of technology-based gait training in an outpatient setting for persons with chronic, severe functional limitations after a neurological event is essential, particularly pertaining to its effects on walking ability and general health as well as assessing economic aspects to provide a basis for the development of an appropriate intervention.

The primary aim of this study was to determine whether a six-months technology-based gait training results in relevant changes of ambulatory function. Secondarily, changes in quality of life will be assessed as well as the willingness to pay (WTP) for technology-based gait training.

## Methods

### Sample

The participants had to fulfill the following inclusion criteria to be eligible for participation in this study: residual gait impairment after neurological event, medical clearance for technology-based gait training, 18–80 years old, no ongoing inpatient rehabilitation, no progressive disease such as multiple sclerosis or Parkinson's disease, and a signed informed written consent. The study was approved by the ethics committee of the canton of Zurich (BASEC-ID: 2020–01481). Participants were recruited at VAMED Rehacenter and through two participating insurance companies.

The sample size calculation was performed based on results from a previous study for walking speed in the 10-meter walk test [23]. This study used patients with a sub-acute stroke. It may be that our participants show slightly diminished improvements as they may also be in a chronic state where smaller improvements can be expected. Using G*Power (G*Power, Kiel, Germany), a sample size of 45 was calculated.

## Study design and intervention

This study was designed as a single-center, longitudinal cohort study. All study procedures took place at the VAMED Rehacenter Zurich Seefeld's former location in Volketswil (Switzerland). Recruitment took place from January 1st 2021 until July 31st 2022. After inclusion, all participants completed six months of technology-based gait training with one or a combination of the following devices: Lokomat® (Hocoma AG, Volketswil, Switzerland), Andago® (Hocoma AG, Volketswil, Switzerland), or C-Mill® (Motek Medical B.V., Houten, The Netherlands) depending on their functional abilities. They were required to complete a minimum of ten trainings per month. The participants were allowed to carry out additional trainings. Each training lasted 60 minutes, including preparatory activities. The trainings were financed jointly by two health insurance companies, the VAMED rehabilitation center and the participants. Due to the nature of the study, no further measures to account for selection bias could be undertaken.

## Outcome measures for functional assessments

The following assessments were performed at baseline (BL), after three and six months of training (M3 and M6, respectively): 10-meter walk test (10MWT, main outcome) [24], 6-minute walk test (6MWT) [25], functional ambulation category (FAC) [26], EQ-5D-3L visual analog scale (VAS) [27], and WHODAS 2.0 [28]. At M3 and M6 the patient global impression of change (PGIC) [29] rating and the number of trainings was obtained. Additionally, the training duration of each training was recorded by the devices. All data was collected using REDCap electronic data capture tools hosted at ZHAW Zurich University of Applied Sciences [30].

## Conjoint analysis

After two weeks of training, the participants completed online a choice-based conjoint analysis survey to compare the attractiveness and WTP of technology-based gait training with other forms of trainings. The Sawtooth Software's online tool, Discover (Sawtooth Software Inc, Provo, Utah), which has already been used for eliciting patient preferences in other studies (e.g., [31]), was used for choice task creation, data selection and data analysis. Patients were asked to select their most preferred treatment alternative from nine screens with three sets of different therapy offers (including the option not to select any of the offers (non-selection option)). These offers enclosed the following attributes: form of therapy (conventional physiotherapy, medical training therapy, technology-based training), supervision ratio (1:1, 1:2, 1:5), length of journey to get to therapy (10, 20, 30 minutes), and cost per hour (CHF 25, 115, 160). This survey was conducted to identify which of these attributes were perceived most important by patients when deciding between different offers (weights of importance) and to assess the relative WTP for the various attributes of each treatment. Market share simulations were used to compare technology-based training with conventional physiotherapy and medical training therapy.

   Patient preference scores for the procedure attributes were calculated based on the response patterns of the included sample. Discover Choice-Based Conjoint estimates preference scores (utilities) for each respondent and each level of your attributes using a statistical estimation approach called Empirical Bayes which was used to generate individual utilities reflecting the individual preferences for each attribute. Raw utilities were then used to run simulations and calculate the market share of technology-based trainings under the Randomized First Choice Rule. For the calculation of the price-market share function, the price of the technology-based offer was varied successively in the market simulation from 25 CHF to 160 CHF, while presenting constant attribute values for the other two forms of therapy (technology-based trainings; supervision ratio (1:2) & 30 min length of journey) to get to therapy reflecting the longer journey to a technology-based training location and the parallel supervision of two patients in this training form. Conventional physiotherapy was defined in the simulation with the attribute values: (conventional physiotherapy, supervision ratio (1:1) & 20 min length of journey to get to therapy), while medical training therapy was presented with the attribute values (medical training; supervision ratio (1:5); 10 min length of journey to get to therapy) reflecting the conditions of therapy access and conditions in the Swiss setting.

Furthermore, the respondents had to state the direct WTP with the question "What price would you pay for technology-based therapy at VAMED Rehacenter per hour as a self-payer in CHF?" using an open response field.

## Statistical analysis

Descriptive statistics (median, interquartile range) were calculated for the three time points (Baseline, M3, M6). To determine changes between these timepoints, Wilcoxon signed rank tests were calculated for all three comparisons individually to account for potentially missing data. As an exploratory analysis, linear mixed-effects models were fit for the outcome measures 10MWT (Equation 1) and 6MWT with random intercepts for the individuals.

$$WT10M_i \sim N\left(\mu, \sigma^2\right)$$

(1)

$$with \; \mu = \alpha_{j[i]} + \beta_1\left(Time\_ord._L\right) + \beta_2\left(Time\_ord._Q\right) + \beta_3(FAC\_ord._L) + \beta_4(FAC\_ord._Q) + \beta_5(FAC\_ord._C) + \beta_6(FAC\_ord_4)$$

$$\alpha_j \sim N\left(\mu_{\alpha j}, \sigma^2_{\alpha j}\right), for \; ID \; j = 1, \ldots, J$$

**Equation 1: Mathematical mixed-effects model for 10MWT with random intercepts**. WT10Mi: observed value of the 10-meter walk test for individual i; N(μ,σ2): normal distribution with mean μ and variance σ2; μ: fixed effects component of the model; αj[i]: random intercept for individual I, drawn from a normal distribution N(μαj,σ2αj) for each individual j; β1: coefficient for the linear term of the time variable Time_ord.L; β2: coefficient for the quadratic term of the time variable Time_ord.Q; β3: coefficient for the linear term of the factor variable FAC_ord.L; β4: coefficient for the quadratic term of the factor variable FAC_ord.Q; β5: coefficient for the cubic term of the factor variable FAC_ord.C; β6: coefficient for the quartic term of the factor variable FAC_ord^4; α~N(μαj,σ2αj): the random intercept αj for each individual j is drawn from a normal distribution with mean μαj and variance σ2αj; ID: identifier for individuals, with j = 1,….,J representing different individuals.

The software R (R Core Team (2024). R: A Language and Environment for Statistical Computing. R Foundation for Statistical Computing, Vienna, Austria. <https://www.R-project.org/>) was used for statistical analysis and visualization. Model fit was assessed using the R package *performance* [32]. Results of the statistical model should be interpreted with caution and in an exploratory manner only due to the small sample size and the nature and goal of our study.

We used the STROBE cohort checklist when writing this report [33]

## Results

While a total of 27 participants (Tables 1 and 2) entered the study and provided data for the conjoint analysis, at baseline, and M3, 20 participants completed six months of training and provided data at timepoint M6. On average, participants reached the target number of 30 trainings in three months.

**Table 1. anthropometric description of participants and number of trainings.**

|  | Median | IQR |
|---|---|---|
| Age [years] | 58.8 | 20.5 |
| Height [cm] | 176 | 12 |
| Mass [kg] | 77 | 20 |
| Time since diagnosis [years] | 3 | 6 |
| Number of trainings per participant between Baseline and M3 | 32 | 4 |
| Number of trainings per participant between M3 and M6 | 31 | 3.5 |

## Functional assessments

Descriptively, all measures indicated an improvement after three months that continued after six months, albeit to a lesser extent (Table 3).

The speed in the 10MWT showed a small increase over the course of the study. It appears that the increase was smaller between M3 and M6 compared to the increase between Baseline and M3 (Table 3, Fig 1). However, since 7 participants did not complete the M6 assessments, these values cannot be compared directly. The change in the speed showed statistical significance for all three comparisons between timepoints (Table 3).

The linear mixed model (Table 4) supports a quadratic relationship of 10MWT with the FAC and a linear relationship with time as ordered factor (see Appendix for example of linear mixed model for 10MWT). Note, that the coefficients cannot be directly interpreted as in the case of an ordinary multiple regression model with numeric predictors since orthogonal polynomials are used as contrasts and the coefficients in the model summary refer to these polynomials.

The results of the 6MWT indicate an improvement from Baseline to M3 and to M6. The improvement between M3 and M6 was smaller compared to the improvement between Baseline and M3; however, again, 7 participants did not complete M6. The change in the distance covered in the 6MWT was statistically significant different for all three comparisons (Table 3, Fig 2). Like the 10MWT, the model for 6MWT (Table 4) supports a quadratic relationship between FAC as ordered factor and 10MWT and a linear relationship for time as ordered factor.

The self-reported outcomes of quality of life as well as the therapist-rated FAC showed no changes. The one exception is the EQ5D VAS that showed an improvement from Baseline to M3 (Table 3).

**Table 2. distribution of diagnosis and sex among participants.**

|  |  | n |
|---|---|---|
| Diagnoses | Stroke | 14 |
|  | SCI | 5 |
|  | Post-Inflammation | 3 |
|  | Tumor | 2 |
|  | Miscellaneous | 3 |
| Sex | Male/ Female | 19/ 8 |

IQR: interquartile range, SCI: spinal cord injury

**Table 3. Descriptive results of assessments at baseline, after three months (M3), and six months (M6) of training.**

|  | Baseline (n=27) | | M3 (n=27) | | M6 (n=20) | | Δ BL-M3 (n=27) | | | Δ M3-M6 (n=20) | | | Δ BL-M6 (n=20) | | |
|---|---|---|---|---|---|---|---|---|---|---|---|---|---|---|---|
|  | Median | IQR | Median | IQR | Median | IQR | Median | IQR | p-value | Median | IQR | p-value | Median | IQR | p-value |
| 10MWT speed [m/s] | 0.46 | 1.10 | 0.54 | 1.10 | 0.57 | 1.11 | 0.09 | 0.15 | 0.0003 | 0.04 | 0.07 | 0.0113 | 0.14 | 0.30 | 0.0023 |
| 6MWT distance [m] | 149 | 404 | 155 | 394 | 159 | 320 | 14 | 31 | 0.0048 | 9 | 26 | 0.0085 | 18 | 64 | 0.0012 |
| FAC | 2 | 2.5 | 3 | 2.5 | 3 | 2.0 | 0 | 0 | 0.5000 | 0 | 0 | 0.5000 | 0 | 0 | 0.2500 |
| EQ-5D-3L VAS | 60 | 20 | 70 | 25 | 67.5 | 33 | 5 | 20 | 0.0113 | 0 | 30 | 0.5862 | 8 | 45 | 0.4065 |
| WHODAS 2.0* [%] | 27 | 34 | 23 | 36 | 27 | 41 | 0 | 14 | 0.4468 | -3 | 10 | 0.2307 | -2 | 18 | 0.2119 |
| PGICS |  |  | 1 | 1 | 2 | 1 |  |  |  |  |  |  |  |  |  |

IQR: interquartile range. *negative change indicates improvement

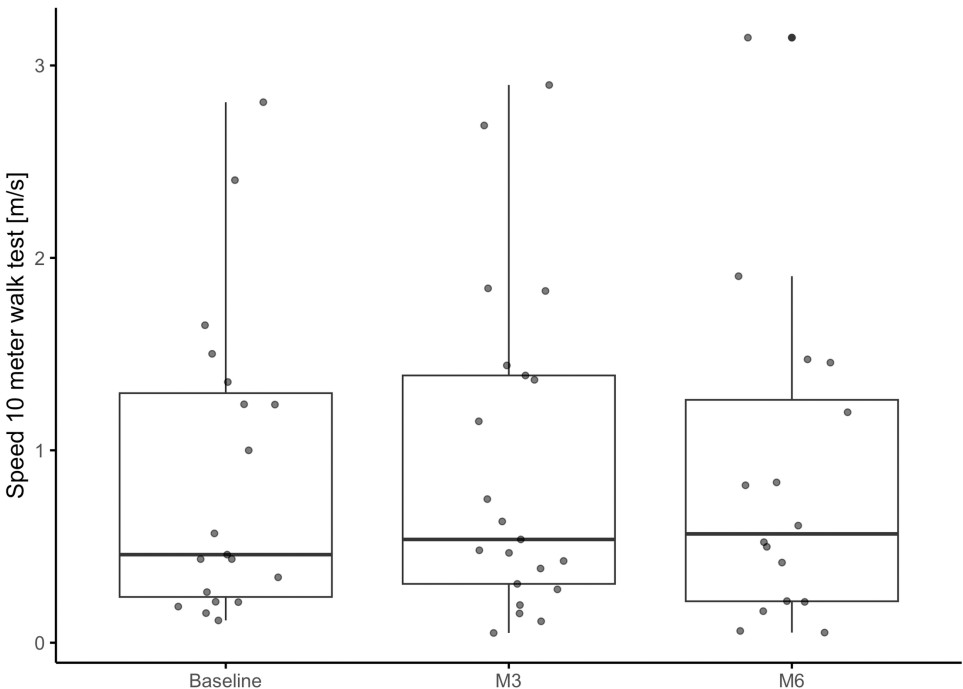

**Fig 1. Boxplots and individual values of the speed of the 10-meter walk test at baseline, M3 and M6 assessments for 21 participants (6 participants were unable to perform this assessment).**

**Table 4. Results from the linear-mixed-effects model for the speed in the 10-meter walk test and the distance in the 6-minute walk test, integrating the ordered factors of time and Functional Ambulation Category (FAC).**

| Speed in 10-meter walk test | | | |
|---|---|---|---|
| Coefficient | Estimate | Std.Error | 95% Confidence Interval |
| (Intercept) | 0.82 | 0.09 | (0.64, 1.01) |
| Time_ord.L | 0.12 | 0.02 | (0.08, 0.16) |
| Time_ord.Q | -0.03 | 0.02 | (-0.07, 0.01) |
| FAC_ord.L | 1.47 | 0.14 | (1.17, 1.76) |
| FAC_ord.Q | 0.66 | 0.15 | (0.36, 0.96) |
| FAC_ord.C | 0.05 | 0.18 | (-0.34, 0.43) |
| FAC_ord^4 | -0.24 | 0.09 | (-0.42, -0.06) |
| **Distance in 6-minute walk test** | | | |
| (Intercept) | 323 | 32 | (255, 391) |
| Time_ord.L | 28 | 6 | (16, 40) |
| Time_ord.Q | -2 | 5 | (-13, 9) |
| FAC_ord.L | 150 | 83 | (-28, 329) |
| FAC_ord.Q | 354 | 75 | (192, 516) |
| FAC_ord.C | -187 | 62 | (-320, -54) |

Time_ord.L: linear coefficient of the variable time; Time_ord.Q: quadratic coefficient of the variable time; FAC_ord.L: linear coefficient of the variable FAC; FAC_ord.Q: quadratic coefficient of the variable FAC; FAC_ord.C: cubic coefficient of the variable FAC; FAC_ord^4: quartic coefficient of the variable FAC.

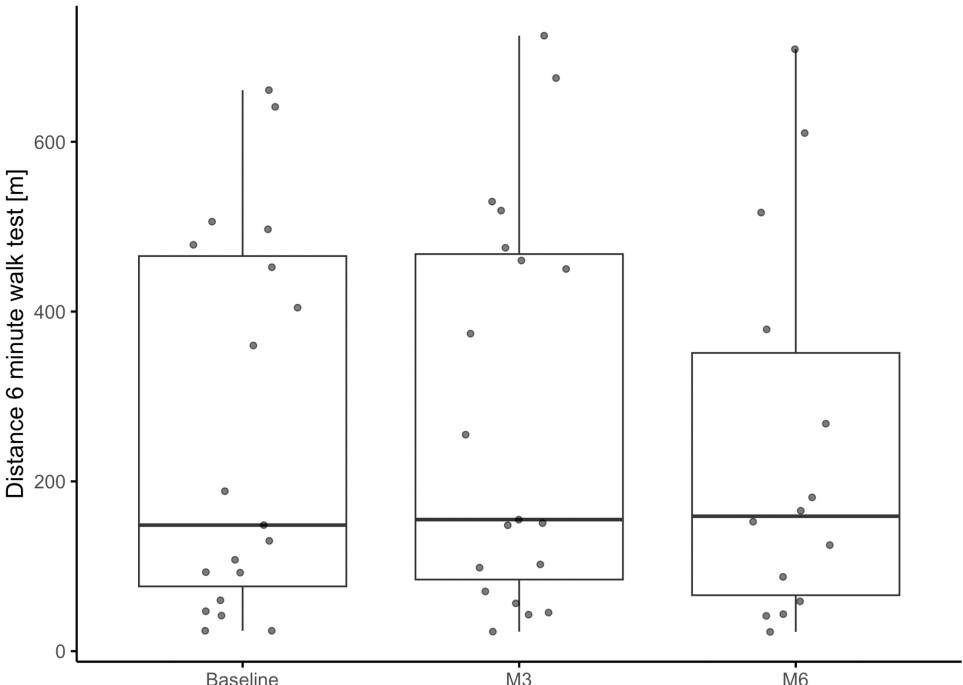

**Fig 2. Boxplots and individual values of the distance covered in the 6-minute walk test at baseline, M3 and M6 assessments for 19 participants (8 participants were unable to perform this assessment).**

## Conjoint analysis and direct willingness to pay (WTP)

For the conjoint-analysis, data from all 27 participants were obtained. It revealed the following importance weights of the categories: cost: 33%, form of therapy: 31%, supervision: 25%, journey; 11%. Market simulations revealed a linear price-market share function for one unit of technology-based trainings: Market share = 82.24–0.364 x price in CHF. This means that the simulation estimated a market share of 48.41% for technology-based training at a price of CHF 105 which is comparable with the actual price of conventional physiotherapy in Switzerland. Higher prices, e.g., CHF 160 for one unit of technology-based training resulted in lower market shares of 18.25%.

The analysis of the direct WTP question revealed a mean WTP of CHF 80 for one unit of technology-based training. (median CHF 80, Min: CHF 0, Max: CHF 160). The market simulation with a price for the technology-based training of CHF 80 per unit estimated a market share of 55% for the technology-based training compared to MTT and conventional physiotherapy.

## Discussion

The goal of this quasi-experimental study was to explore if six months of technology-based gait training results in relevant changes of the ambulatory function and quality of life in people with chronic, neurological impairments. In addition, the WTP for technology-based gait training by participants was analyzed.

### Functional assessments

Descriptively, the 10MWT, an assessment for walking speed over a short distance, showed improvements after three and six months of training. The LMM indicated a linear relationship with Time and 10MWT, but no higher order relationship, supporting the observation of increased speed with training. Comparing Baseline to M6 showed an improvement

of 0.14 m/s. In the literature, a clinically relevant change of 0.16 m/s has been described for people with subacute stroke and mild to severe impairments [34,35]. Consequently, the results indicate a median change that is below clinical relevance. The participants in this study are, however, in a chronic stage. They may require a longer training time to achieve changes compared to patients in an acute or subacute phase. Several studies have shown that the timing for retraining walking after a stroke is a critical factor that influences the effectiveness of rehabilitation [36]. The fact that after six months of training, the improvement was just below the clinically relevant change, may support this assumption. However, this is contrary to a study performed with patients with a spinal cord injury where both acute and chronic patients showed a comparable improvement in the 10MWT [37]. In this study, the patients trained five times per week for 90–120 minutes which is a larger training volume. Also, both the acute and chronic group had a much lower walking speed (before and after training) which limits the comparability of the two studies. It could indicate that patients in a chronic phase with a walking speed in the 10MWT comparable to the researched sample may benefit less from technology-based gait training compared to a sample with reduced function. Further, it is unclear if the clinically relevant change is different in a chronic population. Especially considering that the subjective rating of the PGIC showed an improvement with training, this should be explored in future studies. Even though the linear mixed model suggested a linear relationship between the speed of the 10MWT and the time of training, it cannot be assumed that the improvements will linearly increase when the training is continued indefinitely. More research with longer durations of training but repeated assessments are necessary to identify the dynamic of improvement, i.e., if the improvements in gait speed will plateau or if they will continue and to what extent.

The walking endurance, measured with the 6MWT, indicated an improvement with training, both for the first and the second three-month period as well as over the entire six-months. This was supported by the differences between the timepoints as well as the linear mixed model that showed a linear relationship between the distance covered in the 6MWT and training time. The clinically relevant change for patients with chronic, neurological diagnoses has not been established but comparing the outcomes with different studies, indicates that the improvement observed in our study is at the lower end of the range of clinically relevant changes or lay below that threshold [38–41]. Similarly to the results of the 10MWT, it can be concluded that for patients in a chronic stage, longer duration of training is necessary to achieve relevant changes in walking endurance, but the results give confidence that it may be possible to achieve this threshold. This is in agreement with the study by Zieriacks et al. (2021) who found a larger improvement in the walking performance in an acute population compared to a population with chronic spinal cord injuries [37]. It remains to be determined in future studies if and at what point the improvements in walking function will plateau despite increasingly intensive training plans [42,43].

For both the 10MWT and the 6MWT the linear mixed model indicated a quadratic relationship between the FAC score and the walking speed and walking distance, respectively. The LMM and the raw data support a quadratic relationship between FAC and 10MWT, indicating that patients with a higher FAC may achieve higher improvements in walking compared to patients with lower FAC. This is in contrast to a study investigating the clinical features of patients with subacute stroke in relation to the benefit from robotic walking therapy [44]. In future research, it is important to only include participants with the same FAC score or have sufficiently large samples that allow for the analysis of sub-groups that are separated by FAC. In order to choose patients that may benefit substantially from the intensive training, it is important to know which baseline characteristics influence the outcomes and the FAC appears to be one of the prerequisites. With more research in this area, the technology-based gait training may be prescribed more specifically to patients that may achieve large benefits.

While the FAC needed to be considered in the linear mixed model, the training duration of three and six months did not lead to changes in this score. Comparing Baseline to M3 and M3 to M6, only for two participants per comparison, the therapist's rating of the FAC changed. It may be that the FAC is not sensitive enough to detect changes in walking function in patients with chronic, neurological conditions. A study assessing the predictive validity and responsiveness of the FAC

determined excellent validity and responsiveness; however, in patients 30–60 days after a first-ever stroke [45]. Given the different sample in this study, it remains to be determined if the FAC is an appropriate assessment to assess gait function in patients in a chronic stage and with different neurological diagnoses.

Both participant-rated scores for quality of life (EQ-5D-3L VAS and WHODAS 2.0 overall score) showed (almost) no change related to the training. The only change (EQ-5D-3L VAS baseline vs. M3) did not reach the minimal clinically important difference determined for patients after stroke [46]. This allows the conclusion that the technology-based gait training did not translate into relevant changes in quality of life. This is different from a study that used the Medical Outcomes 36-Item Short Form Health Survey (SF-36) to assess quality of life before and after an eight-week training program in participants with chronic stroke. They found an improvement in all sub-parameters of this scale [47]. Comparing the study samples shows that the present study used participants with less walking function, based on the distance covered in the 6MWT. Overall, there is little information on the effect of technology-based gait training on quality of life in patients in a chronic phase, indicating the need for further research.

The results of the conjoint analysis show that there existed a direct WTP for technology-based training comparable to conventional physiotherapy. Due to the actual low availability of technology-based training, patients will have to drive longer distances to the training location, an attribute which had an importance weight of 10% in the conjoint analysis. Moreover, a 1:1 supervision ratio was valued higher than supervision ratios with more patients per therapist. Both factors, the distance and the lower supervision ratio were negative attribute values of technology-based trainings which consequently may lower the WTP for this training. However, market simulations revealed a 55% market share for technology-based training against conventional physiotherapy and MTT.

## Limitations

One major limitation of this study was the inability to reach the calculated sample size. Consequently, the chosen statistical analysis for the descriptive analysis was adjusted to a non-parametric analysis. This adjustment may have resulted in an underestimation of the potential effects of the training [48]. Future studies need to allow for more time to recruit participants and include multiple institutions for recruitment and training. Especially for study protocols requiring several training sessions per week, a proximity of the training facility to the participant's residence is relevant.

There was a total of seven participants (26%) who dropped out of the study after completing the first three months of training. Consequently, there were no data available for these participants at M6. There were various reasons for dropping out. Some participants reported an improvement that made a continuation of the training obsolete for them. Others had a decrease in health status that was unrelated to the study intervention. And some participants reported that the time commitment of 3 trainings per week was too much for them. The dropout rate was similar to comparable studies [7,47], however, efforts need to be made to decrease the dropout rate in future studies. On the other hand, the adherence to the training was excellent with all participants reaching the targeted minimum of 10 trainings per month. Due to the exploratory nature of our LMM, we refrained from implementing imputation. Furthermore, due to dropout reasons provided, the missingness mechanism seems to be MNAR (missing not at random), i.e., the missingness is associated with the missing value (e.g., improvement which made the training obsolete). MNAR poses problems for LMM and may yield biased results [49].

Further, only the training conducted at the study site was analyzed. It is possible that participants attended prescribed physical therapy or independent trainings outside of the study intervention. The study plan did not exclude any outside training to represent what would typically be done in regular life. But, it cannot be estimated if and to what extent additional training may have affected the results of this study.

The results of the WTP as well as the market share estimations must be interpreted against the fact that the respondents were all active participants of the program and were fully informed about the features and experience with technology-based training. Therefore, the WTP may be lower in the decision-making process of new customers for the service.

## Conclusion

This study, exploring whether a technology-based gait training results in relevant changes of ambulatory function and quality of life as well as the willingness to pay, resulted in improved walking function, however with only small improvements. If the detected changes meet clinically relevant changes for the included population remains to be determined in future research. Combined with the willingness to cover a share of the costs related to the training, it can be concluded that such training fulfills a clinical and market need. Consequently, an offer could be developed to provide people with chronic neurological diagnoses an opportunity for prolonged therapy and training.

## Acknowledgments

The authors would like to thank the participating health insurances Sanitas and SWICA Health Insurance Ltd for their contribution.

## Author contributions

**Conceptualization:** Eveline Graf, Dino De Bon, Daniel Zutter, Florian Liberatore, Markus Wirz.

**Data curation:** Eveline Graf.

**Formal analysis:** Eveline Graf, Jürgen Degenfellner.

**Funding acquisition:** Daniel Zutter, Markus Wirz.

**Investigation:** Eveline Graf, Dino De Bon, Johanna Stahl, Deborah Knechtle.

**Methodology:** Eveline Graf, Dino De Bon, Daniel Zutter, Florian Liberatore, Markus Wirz.

**Project administration:** Eveline Graf, Dino De Bon, Deborah Knechtle.

**Resources:** Dino De Bon, Deborah Knechtle, Daniel Zutter.

**Software:** Eveline Graf, Johanna Stahl, Jürgen Degenfellner.

**Supervision:** Markus Wirz.

**Validation:** Jürgen Degenfellner.

**Visualization:** Eveline Graf, Jürgen Degenfellner.

**Writing – original draft:** Eveline Graf, Johanna Stahl, Jürgen Degenfellner.

**Writing – review & editing:** Eveline Graf, Dino De Bon, Johanna Stahl, Jürgen Degenfellner, Deborah Knechtle, Daniel Zutter, Florian Liberatore, Markus Wirz.

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
