## [Decision Letter · Decision Letter 0]

11 Mar 2025

PONE-D-24-51168Can technology-based gait training result in relevant changes of ambulatory function in people with chronic, neurological diagnoses? A longitudinal, cohort studyPLOS ONE

Dear Dr. Graf,

Thank you for submitting your manuscript to PLOS ONE. After careful consideration, we feel that it has merit but does not fully meet PLOS ONE’s publication criteria as it currently stands. Therefore, we invite you to submit a revised version of the manuscript that addresses the points raised during the review process.

We look forward to receiving your revised manuscript.

Kind regards,

Victor Afamefuna Egwuonwu, PhD

Academic Editor

PLOS ONE

Journal Requirements:

“This research did not receive funding from a funding agency. The cost for the training were partially covered by health insurances Sanitas and SWICA Health Insurance Ltd.”

Additional Editor Comments:

Dear Author,

We have reviewed your manuscript and found it to be well-written and scientifically sound. However, a few minor revisions are required before final acceptance for publication. Please address the following points, highlighted in the attached document, which primarily focus on clarifying certain sections and ensuring consistency with journal style guidelines.

Once these minor adjustments are made, please resubmit your revised manuscript.

Key points to include in the attached document with specific comments:

Clarifications:

Style and formatting:

Ensure consistency with the journal's style guide regarding citations, headings, and presentation of data.

Minor grammatical errors:

If necessary, indicate the exact line number where a change was made.

Important considerations:

Concise feedback:

Focus on the most important areas for improvement.

Reviewers' comments:

Reviewer's Responses to Questions

**Comments to the Author**

1. Is the manuscript technically sound, and do the data support the conclusions?

Reviewer #1: Partly

2. Has the statistical analysis been performed appropriately and rigorously? 

Reviewer #1: I Don't Know

3. Have the authors made all data underlying the findings in their manuscript fully available?

Reviewer #1: Yes

4. Is the manuscript presented in an intelligible fashion and written in standard English?

Reviewer #1: Yes

5. Review Comments to the Author

Reviewer #1: I have attached my comments for your response and this will determine whether the manuscript will be recommended for publication in the journal. Please note that the editor will expect you to provide your feedback in the shortest possible time.

6. PLOS authors have the option to publish the peer review history of their article (what does this mean? ). If published, this will include your full peer review and any attached files.

**Do you want your identity to be public for this peer review?** For information about this choice, including consent withdrawal, please see our Privacy Policy .

Reviewer #1: No

---

## [Author Response · Author response to Decision Letter 1]

11 Apr 2025

Thank you to the reviewer and editor. All comments have been addressed in the "Response to Reviewers" document that has been uploaded.

---

## [Editor Report · Decision Letter 1]

21 Apr 2025

Can technology-based gait training result in relevant changes of ambulatory function in people with chronic, neurological diagnoses? A longitudinal, cohort study

PONE-D-24-51168R1

Dear Dr. Graf,

We’re pleased to inform you that your manuscript has been judged scientifically suitable for publication and will be formally accepted for publication once it meets all outstanding technical requirements.

Kind regards,

Victor Afamefuna Egwuonwu, PhD

Academic Editor

PLOS ONE
---

## [Editor Report · Acceptance letter]

PONE-D-24-51168R1

PLOS ONE

Dear Dr. Graf,

I'm pleased to inform you that your manuscript has been deemed suitable for publication in PLOS ONE. Congratulations! Your manuscript is now being handed over to our production team.

Kind regards,

on behalf of

Dr. Victor Afamefuna Egwuonwu

Academic Editor

PLOS ONE